# Subjective Cognitive Impairment Can Be Detected from the Decline of Complex Cognition: Findings from the Examination of Remedes 4 Alzheimer’s (R4Alz) Structural Validity

**DOI:** 10.3390/brainsci14060548

**Published:** 2024-05-27

**Authors:** Eleni Poptsi, Despina Moraitou, Emmanouil Tsardoulias, Andreas L. Symeonidis, Magda Tsolaki

**Affiliations:** 1Laboratory of Psychology, Department of Cognition, Brain and Behavior, School of Psychology, Aristotle University of Thessaloniki (AUTh), 54124 Thessaloniki, Greece; poptsielena@gmail.com; 2Laboratory of Neurodegenerative Diseases, Center for Interdisciplinary Research and Innovation, Aristotle University of Thessaloniki (CIRI—AUTh), 54124 Thessaloniki, Greece; tsolakim1@gmail.com; 3Greek Association of Alzheimer’s Disease and Related Disorders (GAADRD), Petrou Sindika 13 Str., 54643 Thessaloniki, Greece; 4School of Electrical and Computer Engineering, Faculty of Engineering, Aristotle University of Thessaloniki (AUTh), 54124 Thessaloniki, Greece; etsardou@ece.auth.gr (E.T.); symeonid@ece.auth.gr (A.L.S.)

**Keywords:** fluid intelligence, executive function, working memory, cognitive flexibility

## Abstract

R4Alz is utilized for the early detection of minor neurocognitive disorders. It was designed to assess three main dimensions of cognitive-control abilities: working-memory capacity, attentional control, and executive functioning. Objectives: To reveal the cognitive-control dimensions that can differentiate between adults and older adults with healthy cognition, people with subjective cognitive impairment, and people diagnosed with mild cognitive impairment by examining the factorial structure of the R4Alz tool. Methods: The study comprised 404 participants: (a) healthy adults (n = 192), (b) healthy older adults (n = 29), (c) people with SCI (n = 74), and (d) people diagnosed with MCI (n = 109). The R4Alz battery was administered to all participants, including tests that assess short-term memory storage, information processing, information updating in working memory, and selective, sustained and divided attention), task/rule-switching, inhibitory control, and cognitive flexibility. Results: A two-factorial structural model was confirmed for R4Alz, with the first factor representing “fluid intelligence (FI)” and the second factor reflecting “executive functions (EF)”. Both FI and EFs discriminate among all groups. Conclusions: The R4Alz battery presents sound construct validity, evaluating abilities in FI and EF. Both abilities can differentiate very early cognitive impairment (SCI) from healthy cognitive aging and MCI.

## 1. Introduction

In the past decade, clinicians, and healthcare systems have attempted to find new cost-effective ways to detect cognitive decline early enough during neurodegeneration with the aim of preventing the decline into the clinical stages of dementia, especially Alzheimer’s disease dementia (AD). Besides the fact that mild cognitive impairment (MCI) is considered to be a preclinical stage of major neurocognitive diseases such as dementia because of AD pathology with objective clinical biological and neuropsychological criteria [1] (DSM-5, 2013), there is still a debate regarding whether subjective cognitive impairment (SCI) is a stage of normal aging or a preclinical stage of dementia. SCI is characterized by self-reported experiences of cognitive decline in one or more cognitive domains; however, it does not involve objective evidence of impairment on standardized neuropsychological tests [2,3]. However, it is important to note that SCI should not be diagnosed in patients with emotional factors, such as depression, that could lead to subjective cognitive complaints. 

MCI seems to be a neuropathologically complex syndrome that is difficult to explain or understand within a unique and simple context. MCI is characterized by a variety of pathological changes, including plaque and tangle formation, vascular pathologies, neurochemical deficits, cellular injury, inflammation, oxidative stress, mitochondrial changes, changes in genomic activity, synaptic dysfunction, disturbed protein metabolism, and disrupted metabolic homeostasis [4]. Cognitive changes in MCI have to be evaluated and defined at a level of primary health care and at a very early stage, before the neuropathologic changes of fully developed Alzheimer’s disease appear, for better treatment management to be possible. 

The past years have seen the development and use of several neuropsychological tests such as M@T [5] and MoCA [6,7], batteries such as R4Alz [8], and questionnaires such SCD-Q [9] and the subjective Memory Complaint Scale [10], which aim at the early detection of people with subjective cognitive impairment (SCI) and try to differentiate them from healthy adults and from patients with mild cognitive impairment (MCI) [5,9,11,12]. These tests were developed or used to prove that these three stages of cognitive functioning differ in objective and measurable cognitive parameters [12,13,14,15]. The tools used to prove that these three stages of cognitive functioning differ assess not only the episodic memory deficits associated with AD, but also complex cognition and cognitive-control abilities, as these are impaired early in AD development [12,13,14,15,16,17]. “Complex cognition” is a term used to describe higher-order cognitive processes that involve advanced mental operations such as problem-solving, critical and abstract thinking, planning, reasoning, and decision-making. Complex cognition requires the integration of multiple cognitive skills that are essential for adaptation to novel situations [18,19]. Cognitive control is a part of complex cognition, is a general term used as a synonym for the term “executive function”, and refers to higher-order cognitive abilities [20]. Cognitive control coordinates other inferior cognitive functions, such as working memory, semantic and episodic memory, perceptual attention etc. It is activated when people must cope with unknown conditions and in difficult situations in their lives. Cognitive control allows for dynamic adjustment during varying situations, depending on the goals that a person tries to achieve [21]. For the needs of the present paper, the terms “cognitive control” and “executive function” will be used interchangeably, with the same meaning. In general, great interest regarding cognitive control has emerged in recent decades, with new models constantly developing. In 2000, Miyake at al. developed a theoretical model of the organization of executive functions and their role in cognition called “unity and diversity” [22]. According to this model, there are three important cognitive-control abilities engaged in complex executive tasks: (a) shifting between tasks or mental sets, (b) updating and monitoring of working memory representations, and (c) inhibition of dominant or prepotent responses. The abilities are separable but moderately correlated, indicating both unity and diversity of executive functions [22]. A few years later (2004), Lavie, Hirst, de Fockert, and Viding developed the “load theory”, which suggests that cognitive control and perceptual load are associated with selective attention [23]. Specifically, the authors support the idea that there are two mechanisms that are activated against distractor intrusions: (a) a perceptual-selection mechanism that reduces distractor perception in situations of high perceptual load and (b) a cognitive-control mechanism that acts to ensure that attention is allocated in accordance with current stimulus-processing priorities and thus minimizes intrusions of irrelevant distractors, as long as working memory is available to actively maintain the current priority set (in situations of low working-memory (WM) load) [23] (p. 348). Engle and Kane, the same year (2004), formulated the two-factor theory of cognitive control, which states that WM-capacity tests have a strong relationship with fluid intelligence [24]. They provided evidence that WM reflects the ability to control attention, particularly when other elements of the internal and external environment may capture attention away from the currently relevant test. They proposed a two-factor model by which individual differences in WM capacity lead to performance differences. They argued that WM or executive attention (the ability to maintain stimulus and response elements in active memory, particularly in the presence of events that would capture attention) is, firstly, important for maintaining information in active memory and, secondly, important in the resolution of conflict resulting from competition between task-appropriate responses and prepotent but inappropriate responses. The conflict might also arise from stimulus representations of competing strength [24]. Friedman and Miyake (2017) gave evidence regarding their model of “unity and diversity” of executive functions at the behavioral and genetic levels [25]. As regards the behavioral level, the authors offered evidence that besides the fact that EFs share common functions, there are large differences between patients and inpatients, for whom EF tests show low correlations due to task impurity; therefore, multiple measures are necessary. The inhibition, updating, and shifting functions are combined in the service of more complex EFs such as planning, but they also can be broken down into more basic functions. Furthermore, they offered evidence that EFs are not the same as intelligence and that some EF components differentially relate to intelligence [25]. The summary of the aforementioned theoretical models is presented in Table 1. 

In contrast to the prior belief that memory deficits are the first symptom of AD, there is now evidence that during neurodegeneration, executive functions are impaired early during AD development, even in preclinical stages, including MCI or even SCI [16,17,26]. In fact, there are several studies including biomarkers and neurocognitive data that support the ideas that declines in executive functions are preceded by memory decline in people with SCI [16,27,28] and that cognitive changes may start from cognitive areas follows as the frontal lobe [29,30] and prefrontal cortex [31], areas closely related to executive functions [32].

Specifically, Rabi et al., in their 2020 meta-analysis, having investigated database samples of 2184 adults with amnestic MCI (aMCI) and 3049 controls, showed that generalized inhibition deficit was common among people with aMCI; therefore, they proposed that inhibition tests should be included in the neuropsychological-assessment process [33]. Another recent article, which attempted to determine the nature and extent of minor neuropsychological deficits in people with SCI, analyzed data from 449 participants (healthy controls and SCI) in the DELCODE study and showed that, among other abilities, executive functions, specifically the task-switching ability, were impaired in SCI [13]. Finally, the 2015 study of Smart and Krawitz, which assessed the applicability of the Iowa Gambling Task (IGT) for detecting measurable cognitive differences in SCI, showed that people with SCI score lower than healthy older adults because of difficulties in updating in contexts of uncertainty and doubt [27]. Moreover, there is also evidence from neural data that the underlying decline during healthy aging in dopamine-system projection to the prefrontal cortex can cause cognitive changes in executive functions, namely in working memory, attention, and inhibition [34]. Therefore, it seems crucial to find a way to reveal very early deficits in cognition and especially in higher-order cognitive abilities during the aging process in order to cope with and potentially reverse the first cognitive symptoms of the pathways towards dementia, mainly in AD-related dementia.

Based on this theoretical background, we have developed a tool to capture minor and very early cognitive changes in cognitive control that might exist in the preclinical stages of SCI and MCI, aiming to differentiate healthy cognitive aging from these stages. The design and development of the Remedes for Alzheimer (R4Alz) battery was initiated in 2019 [8] and utilizes a system that does not require the examinee to operate a tablet, PC, or other electronic tools for the implementation of the clinical assessment. This system is REMEDES (Reflexes Measurement Devices), which measures reflexes using visual and auditory triggers (https://lab.issel.ee.auth.gr/remedes/, accessed on 23 May 2024). The system comprises physical, three-dimensional devices (REMEDES pads), controlled from a master device. Every R4Alz battery setup includes seven REMEDES pads that can be activated to provide visual or auditory stimuli, which can be accompanied by a figure/image, depending on the task. For the R4Alz battery, specific figures were printed and placed in front of every pad. The battery’s instructions are both verbal and non-verbal (sketches). According to each task’s instructions, the examinee is asked to do one of the following: (a) deactivate REMEDES pads based on conditions, (b) count specific appearances or sounds, (c) perform both tasks concurrently, or (d) react verbally to specific instructions. Via this system, the proposed battery becomes ecologically valid and alleviates the issues that generate assessment faults for older adults and people with cognitive diseases [8]. The R4Alz was initially designed to mainly assess cognitive control in the following three capacities: (a) working-memory capacity; (b) higher-order abilities involved in attention; and (c) inhibitory control, task-switching, and cognitive flexibility in the form of a combination of these two functions. According to previous pilot studies, the R4Alz battery has a perfect discriminant potential to differentiate people with SCI from healthy controls over 50 years old and healthy controls over 50 years old from people with MCI; it also has excellent discriminant potential between SCD and MCI [12]. 

### The Purpose and the Hypotheses of the Study

Since the R4Alz is a new tool, we considered it important to proceed with the examination of its factorial structure in order to examine the structural validity of the battery. The hypotheses of the study were formulated as follows:

**Hypothesis** **1.**
*Confirmatory factor analysis applied to the data of the R4Alz would indicate three latent factors on which the respective subtests of each of the three tests that constitute the R4Alz battery would load. These subtests have been developed to measure (a) working-memory capacity, (b) attention, and (c) executive functioning in multiple dimensions and/or types, respectively.*


**Hypothesis** **2.**
*The new variables that will be created as the sum of the scores of the subtests loading on each factor could differentiate between healthy cognition in young and older adults and both SCI and MCI.*


## 2. Materials and Methods

### 2.1. Protocol Approval and Participant Consent

All participants were informed orally and in written form of the purpose of the study. They were also informed that their data would be confidentially collected in an electronic database. The participants gave written consent at the time of their first examination, agreeing that their participation was voluntary and that they could withdraw at any time, without the need to give a reason and without cost. The European Union law for personal data protection regulation (GDPR) that has existed since 6 May 2018 was followed. The study’s protocol was approved by the Scientific and Ethics Committee of the Greek Association of Alzheimer’s Disease and Related Disorders (Alzheimer Hellas) (number of decision protocol code 44, approved on 21 November 2018), and followed the principles outlined in the Helsinki Declaration (64th WMA General Assembly) [35].

### 2.2. Participants 

The study sample consisted of 404 volunteers recruited from the broad area of Thessaloniki, Greece. For the purposes of testing the second hypothesis of the study (H2), the sample was divided into four groups. The two groups of (a) cognitively healthy adults (HC adults) and (b) cognitively healthy older adults (HC older adults) consisted of volunteers from the community. The other two groups, SCI and MCI, respectively, were visitors in the Day Care Centers of Alzheimer Hellas (DCCAH) in Thessaloniki during the period March 2019 to April 2023 who visited DCCAH for their yearly medical and psychological check-ups. 

The four groups (HC adults, HC older adults, SCI, MCI), did not significantly differ in gender (χ^2^ (3, 404) = 7.910, *p* > 0.05). However, they differed in age (F(3, 404) = 315.19, *p* < 0.001), and in educational level (grouped as follows: (a) primary education, 0–6 years of schooling; (b) secondary education, 7–12 years of schooling; (c) tertiary education, 13–18 years of schooling; (d) master’s or doctorate’s education, >18 years of schooling; F(3, 404) = 42.609, *p* < 0.001). Regarding age, Scheffe comparisons showed that HC adults differed significantly from HC older adults (I-J = −3.06, *p* < 0.001), from people with SCI (I-J = −2.97, *p* < 0.001), and from the MCI group (I-J = −3.34, *p* < 0.001), as HC adults were markedly younger than HC older adults and people with SCI and MCI. However, HC older adults and the two cognitively pathological groups (SCI, MCI) did not differ in age. As regards educational level, HC adults differed from people with SCI (I-J = 0.70, *p* < 0.001), and MCI (I-J = 0.90, *p* < 0.001), with HC adults being more educated. The HC older-adults group differed from people with MCI (I-J = 0.55, *p* < 0.05), with HC older adults being more educated. Study sample characteristics are presented in Table 2.

### 2.3. Exclusion Criteria

The exclusion criteria were as follows: (a) history of psychiatric illness or affective disorder; (b) substance abuse or alcoholism; (c) history of traumatic brain injury; (d) brain tumor, encephalitis, and neurological disorders; (e) cancer in the last five years, myocardial infarction in the last six months, stroke history; pacemaker; (f) thyroid issues, diabetes; (g) drug treatment with opioids, B12, folate, or thyroid; (h) uncorrected sensory deficits.

All participants underwent an extended neuropsychological assessment for diagnostic reasons [36]. The neuropsychological protocol included the following tests: (a) the Mini Mental State Examination [37,38] (MMSE) and (b) the Montreal Cognitive Assessment Scale [6,7] were used for the assessment of global cognition; (c) the Functional Cognitive Assessment Scale (FUCAS) was used for general functional performance [39]; (d) the Functional Rating Scale for Symptoms of Dementia (FRSSD) was used for ADL (evaluates caregiver’s opinion about the daily functioning of people with MCI) [40]; (e) the Rey Auditory Verbal Learning test (RAVLT), was used for verbal memory [41,42]; (f) the Rey Osterrieth Complex Figure test (ROCFT) was used for visual memory and visual constructive abilities [43]; (g) the Verbal Fluency test (FAS) [44], the Trail Making part B [45,46], and the STROOP test [47] were used for executive function; and (h) the Digit Forwards and Backwards [48] was used for working-memory deficits. The Geriatric Depression Scale (GDS) [49,50] was used in order to exclude patients with depression, whilst Neuropsychiatric Inventory (NPI) was used to exclude neuropsychiatric disorders [51,52]. The Global Deterioration Scale (GDS) [30] was administered to determine a patient’s status regarding the progression of their disease, as follows: stage 1 included people with no cognitive decline and normal functioning; stage 2 included people with SCI, that is, people who expressed worries regarding their symptoms; stage 3 included people with MCI. The first three questions of the subjective Cognitive Decline Questionnaire (SCI-Q) [9] were also used to evaluate complaints regarding cognitive functioning.

### 2.4. Inclusion Criteria

The inclusion criteria regarding HC adults and HC older adults comprised men and women with more than six years of education and without cognitive complaints. Regarding people with SCI, the inclusion criteria were based on the diagnostic criteria proposed by SCI-I Working Group [3] and comprised (a) feelings of worse memory performance not associated with the presence of depressive symptoms; (b) absence of objective cognitive deficits according to the neuropsychological assessment; and (c) disease in stage 2 according to the Global Deterioration Scale [53]. Moreover, all participants had to give positive answers to the first three questions of the subjective Cognitive Decline Questionnaire (SCD-Q) [9] to adhere to the SCI criteria published by Jessen et al. in 2014 [3]. As far as people with MCI are concerned, the inclusion criteria were based on the DSM-5 criteria for mild neurocognitive disorders [1]. Their diagnosis was supported by neurological examination, neuropsychological and neuropsychiatric assessment, neuroimaging (computed tomography or magnetic resonance imaging), and blood tests, and diagnosis was reached by a consensus of specialized health professionals of DCCAH who are considered experts in neurocognitive disorders. The inclusion criteria comprised (a) diagnosis of a minor neurocognitive disorder according to the DSM-5, (b) Mini-Mental State Examination (MMSE) [37,38] total score ≥ 24, (c) disease in stage 3 according to the Global Deterioration Scale [53], and (d) 1.5 standard deviations (SD) below the normal mean according to age and education in at least one cognitive domain according to the utilized neuropsychological tests.

### 2.5. Tools 

#### 2.5.1. The R4Alz Battery 

R4Alz, administered via the REMEDES system (https://lab.issel.ee.auth.gr/remedes/, accessed on 23 May 2024) comprises seven physical, three-dimensional devices (REMEDES pads) that are activated to provide visual and/or auditory stimuli, which can be accompanied by a figure/image, depending on the test. The battery comprises several levels of increasing difficulty that evaluate main cognitive-control abilities [8], aiming to capture each stage of neurodegeneration. The subtests of the R4Alz are described below.

#### 2.5.2. Working Memory Capacity Test

The Short-term Storage Subtest was developed to measure the working-memory component of short-term storage. The examinee must deactivate a series of pads in the correct sequence. The achieved score is the number of pads the examinee managed to correctly remember (the minimum score is two, indicating the worst performance, and the maximum score is seven).

The Processing Subtest measures the working-memory component of information processing. The examinee must deactivate a series of pads in the correct reverse sequence (from the last to the first one). The achieved score is equal to the number of pads the examinee managed to correctly remember (the minimum score is two, indicating the worst performance, and the maximum score is seven). 

The Information Updating Subtest was developed to measure working-memory updating. The task comprises six conditions in which green or/and red pads should be deactivated depending on their color and position (one place on the right side, two places on the right, two places on the right and on the left, etc.) The score indicating the best performance is 14, whilst the theoretical minimum score is −58 (the improbable case in which the examinee deactivates all the erroneous pads and does not deactivate any of the correct ones).

#### 2.5.3. Attention Control Test (ACT)

This test was designed to measure different aspects and levels of attention, from simple ones to aspects requiring inhibitory control or/and set-shifting. It comprises three subtasks and seven levels of difficulty. During the subtests and depending on the conditions, the examinee must deactivate or avoid the deactivation of green and/or red pads, individually or concurrently, as well as count doorbells and/or telephone rings concurrently (or separately) with the pad’s activation. The total score ranges from a minimum of 0, indicating the best performance, to a maximum of 202, indicating the worst performance (the improbable case in which the examinee deactivates all the erroneous pads and does not deactivate the correct ones).

#### 2.5.4. Executive Functioning Test 

The Inhibition and Task/Rule-Switching Subtests 1 and 2 were developed to measure: (1) inhibition and (2) task/rule-switching applications.
(a)The Inhibition part comprises four conditions. In the first, the examinee names animal sketches, whilst in the second, they recognize animal sounds. In the third condition, the examinee names the animal and ignores the heard sound, whilst in the last condition, the examinee names the sound of the animal and disregards the animal image. The total score ranges from a minimum of 0, indicating the best performance, to a maximum of 60, indicating the worst performance.(b)Task/rule-switching part: In the first condition, the examinee names the sound of animals he/she hears until a red pad activation appears. Then, the participant must change the rule and start naming the image of the animal. This procedure occurs several times. In the second condition, the examinee must repeatedly switch between naming the animal sounds and the animal sketches by keeping a specific rule in mind, which is “when the white pad is activated, name the sound you hear, and when the red pad is activated, name the sketch you see”. The total score ranges from a minimum of 0, indicating the best performance, to a maximum of 63, indicating the worst performance. Therefore, the total score, including both executive function subtests (1&2), has a minimum value of 0 and a maximum value of 123, with the score of 123 indicating the worst performance. At this point, it should be mentioned that regarding the first condition of the task/rule-switching subtest, two more variables are calculated: (a) switching errors (SEs), which measures how many times the subject failed in task/rule-switching between sets; and (b) failed sets (FSs), which measures the number of the sets of the test that contain at least one failure. There are five switches among sets; therefore, the best SE score is 0, whilst the worst is 5, indicating five switch errors. Also, there are six sets in total; therefore, the minimum FS score, indicating the best performance, is 0, whilst the maximum score is 6. These two scores are separate variables and do not add up to the subtest’s total score.


The Cognitive Flexibility Subtest was developed to measure the combinatory application of at least two executive functions (inhibition and set-shifting) on a task, reflecting cognitive flexibility. It comprises four conditions in which the examinee must deactivate pads depending on the pictures/sketches he/she sees and the sounds he/she hears, at the same time. The total score has a minimum value of 0, with a maximum score of 56 indicating the worst performance.

### 2.6. Procedure

The administration of the R4Alz battery was performed in two areas: HC adults were assessed on the premises of the Faculty of Engineering, AUTh. The evaluation of the SCI and MCI groups was held in the clinical settings of DCCAH. The assessment took part in well-lit and quiet rooms, away from sounds that could be distractive. The tests were administered by well-trained psychologists and were administered early in the morning to avoid examinee fatigue. The R4Alz tests were conducted in a single session and lasted approximately 90 min.

### 2.7. Statistical Analysis

EQS (version 6.1) statistical software [54] was used to examine the factorial structure of the R4Alz battery. Regarding the confirmation of a CFA model, a non-significant level of goodness-of-fit index χ^2^, that is, *p* > 0.05, is indicative of a good fit of the model to the data. In addition, when the value of root mean square error of approximation (RMSEA) is <0.05, it is also an indication of the good fit of the model to the data. RMSEA values ranging from 0.06 to 0.08 indicate a reasonable and therefore acceptable approximation error. Comparative fit index (CFI) examines whether the data fits a hypothesized model compared to the basic model. Values greater than 0.90 indicate an adequate fit of the model to the data, whereas values close to 1.00 indicate a good fit [36]. The standardized root mean square (standardized RMS), which represents the square root of the difference between the residuals of the sample covariance matrix and the residuals of the hypothesized model was also utilized. Moreover, to improve the model fit, we examined the modification indices, namely the Wald and the Lagrange tests, which represent statistics frequently used to identify focal areas of a misfit in a CFA solution [55]. IBM SPSS Statistics for Windows (version 23.0) was also used (IBM Corp, Armonk, NY, USA).

Multivariate analysis of variance (MANOVA) was applied to the data in a second step, after new variables were created using the sums of the scores of the subtests found to load on each of the underlying factors of the R4Alz. Demographic characteristics and the diagnostic group were set in the analysis as the independent variables, and the new variables were set as the dependent ones. The Scheffe test was used for post hoc multiple comparisons, and it was chosen because it tests all possible comparisons, is robust in relation to non-normality, and provides maximum protection against type I error [56].

## 3. Results

### 3.1. The Factor Structure of the R4Alz

As aforementioned, the R4Alz battery was developed as a tool for measuring working-memory capacity, attention, and executive functioning in their main dimensions and/or subtypes. Hence, the first structural model we tried to confirm was a two-factorial model, with working-memory capacity and executive functioning as the two latent variables on which their respective observed variables’ (subtests) scores loaded, including a freely covarying variable of attention (since attention was represented by only one observed variable, the total score for all its subtests). This model was not confirmed. Given that all these abilities can be considered as constituents of cognitive control, we subsequently tried to confirm a unifactorial model with one latent factor representing the general construct of cognitive control. Again, the model was not confirmed. Hence, based on the modification indices, we proceeded with a two-factorial model in which the factors were allowed to freely covary. The fit indices of this model indicated an almost excellent fit (χ^2^(13, 404) = 19.071, *p* = 0.12, CFI = 0.99, SRMR = 0.01, RMSEA = 0.03 (90% CI: 0.00–0.06)). Based on the inspection of factor loadings, the first factor was considered to reflect a broad cognitive dimension of “fluid intelligence (FI)”, since (a) working memory storage, processing, and updating; (b) attention; and (c) cognitive flexibility as a combinatory application of inhibitory control and set-shifting were found to load on this factor, having high loading weights (>|0.50|). The second factor was considered to represent “executive functioning (EF)”, since the three variables related to the application of inhibition or of set-shifting on the task were found to load on this factor with very high loadings (>|0.80|). The two factors were found to be highly covariant. Moreover, correlations between all subtests’ scores loaded on the first factor and the total score of the incorrect responses in the subtests of inhibition and task-switching, which cannot be explained by the latent factors, were also added to increase the two-factorial model’s fit to an excellent level. After these steps, the Wald Test and the Lagrange Multiplier Test did not support the elimination or the addition of other parameters (Figure 1).

### 3.2. Calculation of Total Scores for the Two Factors

To examine whether the two battery’s factors can discriminate among healthy cognition, SCI, and MCI, two total scores were calculated by using the subtasks’ scores loaded in each factor. To compute the two total scores, we normalized the sub-scores of each factor to obtain a common range. The easiest approach was to normalize all $X_{i}$ values using the min-max normalization technique, as follows:$${X^{\prime }}_{i}=\frac{X_{i}-min(X)}{\max\left(X\right)-min(X)}$$

This step generated an $X^{\prime }$ set comprising all the normalized variables that present a significant statistical difference, with values in the range [0, 1]. The next step was the scoring mathematical formulation, for which we created a mathematical formula that combined all normalized scores, so as to produce two total scores. The approach of summing all involved scores was adopted. Since all scores are bounded to [0, 1], all variables contribute with the same weight/importance towards generating the two total scores. Since each score represents errors, the best performance is represented by the smallest value.

Specifically, the first total score, F1, concerns the variables that load on Fluid Intelligence, which are the Cognitive Flexibility test (CFt), the Short-term Storage subtest (StSs), the Processing subtest (Ps), the Information Updating subtest (IUs) and the Attentional Control test (ACt). Based on the methodology described above, for each of these scores, min-max normalization was applied, resulting in the following variables that are bound to [0, 1]: $X_{CFt}^{\prime },\;X_{StSs}^{\prime },\;X_{Ps}^{\prime },\;X_{IUs}^{\prime },\;X_{ACt}^{\prime }$. The F1 total score is then formulated as follows: $F1=X_{CFt}^{\prime }+X_{StSs}^{\prime }+X_{Ps}^{\prime }+X_{IUs}^{\prime }+X_{ACt}^{\prime }$, with the score having a minimum value of 0 and a maximum of 5, since five normalized variables are summed.

Regarding the second total score, F2, which relates to Executive Function, the variables that contribute are the Inhibition and Task/Rule-switching subtests 1&2 (ITR), the Inhibition and Task/Rule-switching subtests 1&2 Switch Errors (ITRSE) and the Inhibition and Task/Rule-switching subtests 1&2 Failed Sets (ITRFS). Based on the methodology described above, for each of these scores, min-max normalization was applied, resulting in the following variables that are bound to [0, 1]: $X_{ITR}^{\prime },\;X_{ITRSE}^{\prime },\;X_{ITRFS}^{\prime }$. The F2 total score is then formulated as follows: $F2=X_{ITR}^{\prime }+X_{ITRSE}^{\prime }+X_{ITRFS}^{\prime }$, having a minimum value of 0 and a maximum of 3, since three normalized variables are summed.

### 3.3. Group Differences in FI and EFs 

The subsequent MANOVA showed that only the diagnostic group had a significant effect on the new variables (F(3, 404) = 33.70, *p* < 0.001, η^2^ = 0.21. Diagnostic group significantly affected Fluid Intelligence (FI) performance (F(3, 404) = 56.08, *p* < 0.001, η^2^ = 0.30. Scheffe post hoc comparisons showed that FI discriminates SCI from HC adults (I-J = 0.86, *p* < 0.001), from HC older adults (I-J = 0.38, *p* < 0.05), and from MCI (I-J = −0.47, *p* < 0.00). It also affected Executive Functioning (EF) performance (F(3, 404) = 58.57, *p* < 0.001, η^2^ = 0.31. Scheffe post hoc comparisons showed that EF discriminates SCI from HC adults (I-J = 0.68, *p* = 0.001), from HC older adults (I-J = 0.51 *p* < 0.001, and from MCI (I-J = −0.57, *p* < 0.001; Figure 2).

## 4. Discussion

### 4.1. Construct Validity of the R4Alz 

R4Alz was designed for assessing the main dimensions of cognitive control. Based on this, our initial hypothesis was that CFA would confirm a respective model. However, Hypothesis 1 was not confirmed. Instead of the hypothesized structure, a two-factor model with one factor reflecting Fluid Intelligence and another one representing Executive Functioning was confirmed. This CFA model has excellent fit to the data. Hence, the R4Alz tool is an instrument with well-documented construct validity. The Fluid Intelligence factor was found to be loaded by the working-memory capacity, total attention score and the cognitive flexibility total score (CFT) subtests. Fluid intelligence is a wide umbrella concept that encompasses the biologically determined dimensions of intelligence (cognitive mechanics) [57,58], and it is usually used when people are facing a relatively new task that cannot be performed using crystallized intelligence. According to the literature, FI is reflected in cognitive processes such as forming, recognizing, making inferences regarding corresponding implications, problem-solving, reasoning [59], processing and updating information in working memory [60], and cognitive flexibility [61]. Moreover, attentional abilities, especially the upper-level aspects of attention, have been also found to closely relate to FI [62]. Furthermore, according to a recently developed theoretical model, fluid intelligence is almost synonymous with cognitive flexibility. According to this view, fluid intelligence can be seen as an ability that is mainly present in novel and complex situations in which there are no overlearned schemata (crystallized intelligence/pragmatics) and people need to adopt flexible ways of coping [63]. Hence, based on the findings of the present study, it seems that working-memory capacity, attention, and the ability to combine executive functions, all together (and not every cognitive function separately), constitute a strong underlying “mechanism” of cognition that is needed to successfully handle issues of at least relatively high difficulty. Moreover, this “mechanism” and each of its constituents are related to inhibitory control and set-shifting functions. The finding of these double-level relationships (at the level of the factors and at the level of specific indicators’ relationships; see Figure 1) indicates the important, multipath, and discrete role that the specific executive functions can play in supporting the broader fluid-intelligence ability.

The Executive Function factor was found to be loaded by all scores of the subtests of inhibition and task-switching functions. According to the classic model of Miyake et al., published in 2000 [22], inhibition and task/rule-switching are two of the most important cognitive-control abilities engaged in executive tests. Regarding the present study, as already stated, their importance in supporting performance on broader intelligence tests is surely confirmed from the finding of their double-level relationships with fluid intelligence. Nevertheless, on the other hand, it is extremely interesting that inhibition and task-switching scores, but not the score of their combination, constitute a discrete dimension of the cognitive system, as supported from the fact that they load on a second factor and not on FI. This finding could be explained by technical factors, since three different scores were created to measure the application of each of these executive functions on different tasks. However, it could also show that the main executive functions are distinct from general fluid intelligence, potentially because they are supported by somewhat different brain networks, at least at a stricter prefrontal-cortex-involvement level. In this light, the loading of the cognitive-flexibility subtest on F1 could be explained by the potential additional requirements for working memory and attention resources of the more complex tasks.

### 4.2. R4Alz’s Discriminant Potential 

The second hypothesis of the study was that the new variables created based on the factor loadings on the two factors could differentiate people with SCI from cognitively healthy adults and older adults, as well as from people diagnosed with MCI. Hypothesis 2 was confirmed. According to the findings, the total score of fluid intelligence differentiates people with SCI from cognitively healthy adults and older adults. According to the literature, people with SCI, compared to healthy controls, have reduced short-term storage capacity [64,65], difficulties in updating [27], and a reduced attentional network [66]. Hence, it seems that fluid intelligence, as assessed by the R4Alz, can very well show these decrements and differentiate healthy cognition from cognition with subtle impairment that the available neuropsychological batteries cannot capture. As regards the differentiation between SCI and MCI, previous studies have shown that changes in specific types of attention (sustained attention and vigilance) exist in people with SCI, and attention abilities are useful to the differentiation of these groups, since MCI people have worse performance on them [67]. Attention difficulties [68] and WM processes such as updating and monitoring are also impaired in people with MCI and differentiate them from healthy controls [69].

As far as executive functioning is concerned, regarding SCI, Rao et al., in 2023 [70], indicated that people with SCI in comparison to healthy controls, showed a reduced ability to inhibit conflicting responses, whilst Fonseca et al. (2015) [71] showed that people with SCI have low performance in switching tests. Both studies support the idea that inhibition, as well as switching, are early markers or predictors of cognitive decline and more severe cognitive impairments that may lead to MCI [71]. Moreover, several studies have shown that people with MCI cope with specific inhibition problems such as intrusion errors in the proactive interference tasks, as compared to healthy controls [72]. Both SCI and MCI have difficulties in task/rule-switching, however patients with MCI perform worse than SCI, by making more perseverative errors [73]. Hence, based on the finding of group differences, it seems that executive functioning as assessed by the R4Alz can show these decrements as well, and differentiate subtle impairment that the available neuropsychological batteries cannot capture both from healthy cognition and cognition with more severe impairment (mild cognitive impairment), which is obvious in the neuropsychological assessments.

### 4.3. Limitations and Future Work

One of the limitations of the present study concerns the length of the R4Alz battery’s administration, which requires almost one and a half to two hours. Another limitation is that no correlation between biological indicators (e.g., APOE status) and performance in the R4Alz battery was assessed.

Moreover, since in this study, no cut-off scores are provided, our future goal is to provide to the readers and to the clinical healthcare providers the cut-off scores of the battery that can discriminate clinical samples from healthy ones.

Finally, it must be noted that via the current study, we provide to the healthcare professionals more information regarding the abilities that the battery requires; therefore, it can be used in any clinical setting for the assessment of higher-order cognitive abilities, which are important for the clinical diagnosis of neurocognitive diseases.

### 4.4. Clinical Implication of the R4Alz Battery and Innovative Contributions 

The diagnosis of major neurocognitive disorders, especially AD, requires clinicians to take into account many features and markers, both biological and neuropsychological. At the neuropsychological level, the phenotype of major neurocognitive diseases includes the presence of early and significant loss of cognitive abilities (not only of episodic memory), as well as functional impairment, with or without other behavioral changes. At the biological level, lower β-amyloid protein levels, β-amyloid deposition, and CSF measures of elevated total tau or phosphorylated-tau are common biological markers that are utilized in research and routine diagnostical practice. Therefore, several clinical and medical measures are used daily in order to diagnose neurodegeneration and neurocognitive diseases. However, the target here is to provide a new tool that can help in early diagnosis, before the onset of major neurocognitive disease. 

In this vein, the R4Alz is a novel tool that does not focus on the diagnosis of major neurocognitive diseases; instead, it was designed as a screening test to capture the very early and almost subtle changes in executive functions with which mild neurodegenerative disorders, such as SCI, seem to begin. As a consequence, the battery focuses on primary healthcare services, where the very early cognitive symptoms can be diagnosed to target the implementation of preventive strategies, such as cognitive training. According to the current study and to studies of the previous tool [12,74], it seems that the battery has good psychometric abilities, with excellent discriminant potential between healthy cognition and mild neurocognitive disorders; it evaluates complex cognition and is not affected by age, education, or cultural effects. Therefore, it can be a useful screening tool that can be utilized in primary healthcare services, since other screening cognitive tasks used in primary care, such as M@T, the MMSE, the clock-drawing test CDT, and the attention/executive function (AQT), cannot differentiate SCD from MCI with high accuracy [5,75]. Finally, we consider that the final cut-off scores of the battery and the provision of the test in an electronic format will help clinicians achieve early diagnosis of minor neurocognitive disorders. 

## 5. Conclusions

As a result, both fluid intelligence and executive functioning, as assessed by the R4Alz battery, can objectively indicate the very early, subtle changes in cognition usually experienced by older people who express cognitive complaints, but whose cognition is evaluated by the neuropsychological tests as “normal” (subjective cognitive impairment). Moreover, both abilities can differentiate these people (SCI) from cognitively healthy adults and older adults, which is extremely important given the insufficiency of the existing batteries to differentiate very early cognitive impairment from healthy cognitive aging. Finally, the same abilities measured by the R4Alz can differentiate subtle impairment from a more advanced decline, such as MCI.

## Figures and Tables

**Figure 1 brainsci-14-00548-f001:**
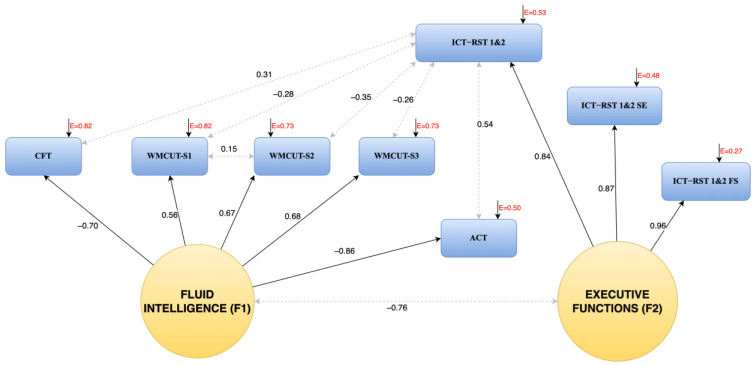
The two-factor structure of the R4Alz; Black arrows indicate the loading of a test in a factor, whereas gray dashed arrows indicate intercorrelation between tests; Factor loadings are statistically significant at *p* < 0.05; E = measurement error.

**Figure 2 brainsci-14-00548-f002:**
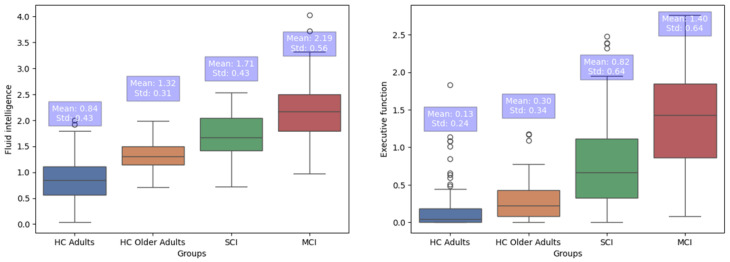
The scores of the four diagnostic groups in Fluid Intelligence and Executive Functioning, presented as boxplots. Abbreviations: HC: cognitively healthy; SCI: people with subjective cognitive impairment; MCI: people with mild cognitive impairment.

**Table 1 brainsci-14-00548-t001:** Summary of the theoretical models concerning cognitive-control abilities.

The “unity and diversity” model [22]	Three cognitive-control abilities engaged in complex executive tasks: (a) shifting between tasks or mental sets (b) updating and monitoring of working-memory representations (c) inhibition of dominant or prepotent responses The abilities are separable but moderately correlated, indicating both unity and diversity of executive functions.
The “load theory” [23]	Cognitive control & perceptual load are associated with selective attention. Two mechanisms are activated against distractor intrusions: (a) a perceptual-selection mechanism that reduces distractor perception in situations of high perceptual load (b) a cognitive-control mechanism that acts to ensure that attention is allocated in accordance with current stimulus-processing priorities and minimizes intrusions of irrelevant distractors as long as working memory is available to actively maintain the current priority set
The “two-factor” theory of cognitive control [24]	Tests of working memory (WM) and fluid intelligence are related. WM reflects the ability to control attention, particularly when other elements of the internal and external environment could capture attention away from the currently relevant test. Individual differences in WM capacity lead to performance differences. WM or executive attention is important for maintaining information in active memory and secondly, is important in the resolution of conflict resulting from competition between task-appropriate responses and prepotent but inappropriate responses.
The “unity and diversity” model of executive functions in a behavioral and a genetic level [25]	At a behavioral level Executive functions (EFs) share common functions but have large differences between patients and inpatients. EF tests show low correlations due to task impurity; therefore, multiple measures are necessary. Inhibition, updating, and shifting are combined in the service of more complex EFs such as planning. EFs can be broken down into more basic functions. EFs are not the same as intelligence. Some EF components differentially relate to intelligence.

**Table 2 brainsci-14-00548-t002:** Study sample demographic characteristics (n = 404).

Diagnostic Groups				
Characteristics	HC Adults (n = 192)	HC Older Adults (n = 29)	SCI (n = 74)	MCI (n = 109)
Age M (SD)	36.95 (12.88)	66.65 (4.76)	65.90 (7.75)	69.46 (7.87)
Gender (Male/Female)	70 M/122 F	11 M/18 F	16 M/58 F	28 M/81 F
Education M (SD)	16.54 (2.70)	15.41 (2.84)	13.56 (4.17)	12.59 (4.13)
MoCA M (SD)	28.20 (1.68)	28.24 (1.20)	26.96 (2.03)	24.62 (3.12)

Abbreviations: HC: Cognitively Healthy; SCI: People with subjective cognitive impairment; MCI: People with mild cognitive impairment, MoCA: Montreal Cognitive Assessment.

## Data Availability

The data are unavailable due to privacy reasons and participant’s ethical restrictions, since the study’s participants did not give informed consent to data sharing.

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
