# Peer review of "Subjective Cognitive Impairment Can Be Detected from the Decline of Complex Cognition: Findings from the Examination of Remedes 4 Alzheimer’s (R4Alz) Structural Validity"

_brainsci, 2024, doi:10.3390/brainsci14060548_

Round 1
Reviewer 1 Report
Comments and Suggestions for Authors
First of all, I would like to congratulate to the authors for the topic they have selected. I have some minor concerns that I am going to explain:
1) Rewrite the Abstract: First explain the idea to the necessity of correct diagnosis of subjective cognitive decline, rather than explaining directly the tool that you propose, as well as to mention what's the term complex cognition... Add info about recruitment strategy and more detailed information about inclusion criteria.
2) Introduction: explain better the terms SCI vs MCI. Differentiate between syndromic and etiological diagnosis. Explain that currently diagnosis of AD should be done in MCI stage, and explain its biological definition.
In turn, summarize the terms of complex cognition and concept of control of cognition. Consider to add a figure to explain the concept and a table to summarize the evidence that is available until present time.
3) Methods: explain better inclusion criteria. Moca is the only cognitive global examination test? If not (I hope so) add the complete NPS protocol. In turn, the MCI have etiological diagnosis of AD?
Do you have information about other health comobidities? For instance, CAIDE dementia risk score for all the categories, drugs used (especial focus in psychodrugs), AOS, epilepsy, ... and findings in MRI (Fazekas scale, ATM scale...).
3) Discussion: justify the need of this tool in current clinical practice where syndromic diagnosis is far from the main objetive, and definition is more complex biological-clinical. Make a summary table comparing the improvements of the suggest tool compared to others accesible in the market, also include in this table aspects to improve, and your suggestions to when and how to incorporate for clinical practice.
Author Response
We thank the reviewer for their to-the-point comments regarding our publication. We have tried our best to properly fix the issues described. Below, you may find the response to each of the comments:
Comment #1.1: Rewrite the Abstract: First explain the idea to the necessity of correct diagnosis of subjective cognitive decline, rather than explaining directly the tool that you propose, as well as to mention what's the term complex cognition... Add info about recruitment strategy and more detailed information about inclusion criteria.
Answer #1.1: We thank the reviewer for their suggestion. Indeed it is important to give the proper theoretical background behind the early diagnosis of neurocognitive disorders, as well as the recruitment strategy and the inclusion criteria. Unfortunately, we are bound to have 200 words maximum, as the MDPI template suggests. Therefore, we believe that the requested information is well presented in the rest of the corpus.
Comment #1.2: Introduction: explain better the terms SCI vs MCI. Differentiate between syndromic and etiological diagnosis. Explain that currently diagnosis of AD should be done in MCI stage, and explain its biological definition.
Answer #1.2: We thank the reviewer for their comment. We attempted to better explain and resolve the required issues. The information is provided in the introduction text, lines 40-63.
Comment #1.3: In turn, summarize the terms of complex cognition and concept of control of cognition. Consider to add a figure to explain the concept and a table to summarize the evidence that is available until present time.
Answer #1.3: Concerning the complex cognition term, we have added the definition in lines 70-74. Concerning the term control of cognition (or cognitive control), its brief description (summary) already exists in lines 70 to 77. Whether the reviewer needs extra information on the subject (or less, more summarized information), please let us know. Furthermore, we added a table to summarize the theoretical evidence available until the present time.
Comment #1.4: Methods: explain better inclusion criteria. Moca is the only cognitive global examination test? If not (I hope so) add the complete NPS protocol. In turn, the MCI have etiological diagnosis of AD?
Answer #1.4: We thank the reviewer for a change to enhance our document. No, MoCa was not the only test used. We have used an extensive neuropsychological battery, we had added as a reference, nevertheless, we altered the text and presented the full neuropsychological protocol we used. The MCI group had the diagnosis of MCI of any etiology. As regards the inclusion criteria, we consider that they have been explained adequately, in a way that we have followed in several other studies. Unfortunately, the reviewer’s comment is a bit vague, therefore, whether the reviewer considers that we have to add further information, please let us know what this information is.
Comment #1.5: Do you have information about other health comobidities? For instance, CAIDE dementia risk score for all the categories, drugs used (especial focus in psychodrugs), AOS, epilepsy, ... and findings in MRI (Fazekas scale, ATM scale...).
Answer #1.5: We thank the reviewer for their question. Unfortunately no extra information about other comorbidities are available, except from the commonly accepted comorbidities or disorders that can lead to cognitive deficits, described in the exclusion criteria: (a) history of psychiatric illness or affective disorder; (b) substance abuse or alcoholism; (c) history of traumatic brain injury; (d) brain tumor, encephalitis, and neurological disorders; (e) cancer in the last 5 years, myocardial infarction in the last 6 months, stroke history; pacemaker; (f) thyroid issues, diabetes; (g) drug treatment with opioids, B12, folate, or thyroid; (h) uncorrected sensory deficits.
Comment #1.6: Discussion: justify the need of this tool in current clinical practice where syndromic diagnosis is far from the main objetive, and definition is more complex biological-clinical. Make a summary table comparing the improvements of the suggest tool compared to others accesible in the market, also include in this table aspects to improve, and your suggestions to when and how to incorporate for clinical practice.
Answer #1.6: We thank the reviewer for the comment. Even though we didn’t use a table, we added to the discussion section two more paragraphs under the title “Clinical implication of the R4Alz battery and innovative contributions” in which we followed your suggestions.
Reviewer 2 Report
Comments and Suggestions for Authors
The paper is well-written and provides a comprehensive account of the subject matter. I have no significant objections. Nevertheless, I have a few minor suggestions for improving the manuscript's presentation.
1. Introduction
1.1 It would be beneficial to provide a brief description of the cognitive test batteries/tests utilized between lines 40 and 44, as this would provide a more comprehensive understanding of the context for the reader.
1.2 Including a section in the introduction that describes studies conducted with R4Alz would be beneficial. The current presentation of the main instrument, R4Alz, does not sufficiently highlight its importance.
2. Materials and Methods
2.6.1 The R4Alz battery
In each of the sub-items, citations should be included where the test has been used previously. It is necessary to cite the Attention Control Test (ACT) and the Executive Functioning Test in the appropriate sections.
Comments on the Quality of English LanguageIn my opinion, there is a need for a more technical English revision of the manuscript.
Author Response
We thank the reviewer for their to-the-point comments regarding our publication. We have tried our best to properly fix the issues described. Below, you may find the response to each of the comments:
Comment #2.1: It would be beneficial to provide a brief description of the cognitive test batteries/tests utilized between lines 40 and 44, as this would provide a more comprehensive understanding of the context for the reader.
Answer #2.1: Thank you for the comment. We added the names of the most famous tests/batteries/questionnaires in the text.
Comment #2.2: Including a section in the introduction that describes studies conducted with R4Alz would be beneficial. The current presentation of the main instrument, R4Alz, does not sufficiently highlight its importance.
Answer #2.2: Since R4Alz is a new neuropsychological battery, no finished studies exist thus far, for presenting them i the introduction. Nevertheless, the tool is currently being used in the following studies
a) Cognitive deficits in people with cardiological problems
- b) Cognitive deficits in people with frontotemporal dementia
- c) Updating of working memory in healthy older adults with sleep disorders
- d) Cognitive deficits in people with burn-out syndrome
Comment #2.3: The R4Alz battery: In each of the sub-items, citations should be included where the test has been used previously. It is necessary to cite the Attention Control Test (ACT) and the Executive Functioning Test in the appropriate sections.
Answer # 2.3: As mentioned in the previous comment, no finished studies exist to add more citations regarding the test. Furthermore, the ACT and the EFT are parts of the R4Alz battery. These are no extra or separate tests. So, the citation is the R4Alz battery (Poptsi et al., 2019), that already exists.
Reviewer 3 Report
Comments and Suggestions for Authors
This paper confirmed that the three latent factors comprising the R4Alz battery cannot successfully load the respective subtests of each of the three tests and then developed new variables to differentiate healthy cognition in young and older adults, SCI and MCI between each other, which have potential implications in the clinical diagnosis of neurocognitive diseases. There are some problems that would be better if they could be solved:
1. Additional detailed information about R4Alz can be included in the background introduction.
2. The confirmation process of Hypothesis 1 lacks figure or table for demonstration.
3. The creation process of the two new variables is not explained in the text.
4. The author need to highlight this paper's innovative contributions.
Comments on the Quality of English LanguageNone.
Author Response
We thank the reviewer for their to-the-point comments regarding our publication. We have tried our best to properly fix the issues described. Below, you may find the response to each of the comments:
Comment #3.1: Additional detailed information about R4Alz can be included in the background introduction.
Answer #3.1: Thank the reviewer for the comment. We added extra information regarding the battery. However, it is also described extensively in the previously published papers and in the Method section.
Comment #3.2: The confirmation process of Hypothesis 1 lacks figure or table for demonstration.
Answer # 3.2: According to hypothesis 1 the CFA would indicate three latent factors on which could load the respective subtests of each of the three tests that constitute the R4Alz battery and have been developed to measure (a) working memory capacity, (b) attention, and (c) executive functioning, in multiple dimensions and/or types of them, respectively. However, against our hypothesis, the battery loaded in two factors (FI and EF). The confirmation of the hypothesis is the figure 1.
Comment #3.3: The creation process of the two new variables is not explained in the text.
Answer #3.3: We thank the reviewer for highlighting this issue. We are describing the creation process of the two new variables in section 3.2. Nevertheless, based on your comment we understood that the described process is not clear enough, therefore we extended this section.
Comment #3.4: The author need to highlight this paper's innovative contributions.
Answer #3.4: We thank you for the comment. Indeed we added to the discussion section two more paragraphs under the title “Clinical implication of the R4Alz battery and innovative contributions” that we think highlight the paper and the R4Alz’s innovative contributions.
Round 2
Reviewer 3 Report
Comments and Suggestions for Authors
No additional comments.